# AlignNet: Self-supervised alignment module

## Abstract

The natural world consists of objects that we perceive as persistent in space and time, even though these objects appear, disappear and reappear in our field of view as we move. This can be attributed to our notion of object persistence – our knowledge that objects typically continue to exist, even if we can no longer see them – and our ability to track objects. Drawing inspiration from the psychology literature on 'sticky indices', we propose the AlignNet, a model that learns to assign unique indices to new objects when they first appear and reassign the index to subsequent instances of that object. By introducing a persistent object-based memory, the AlignNet may be used to keep track of objects across time, even if they disappear and reappear later. We implement the AlignNet as a graph network applied to a bipartite graph, in which the input nodes are objects from two *sets* that we wish to align. The network is trained to predict the edges which connect two instances of the same object across sets. The model is also capable of identifying when there are no matches and dealing with these cases. We perform experiments to show the model's ability to deal with the appearance, disappearance and reappearance of objects. Additionally, we demonstrate how a persistent object-based memory can help solve question-answering problems in a partially observable environment.

## 1 Introduction

Despite head and body movements and the motion of objects, we perceive the world and the objects in it as being spatially stable (Pylyshyn, 1989). There are two factors that contribute to this stability: Firstly, we typically know where objects are even if we can't see them anymore. Secondly, we are able to keep track of objects over time, even when they disappear and reappear after some delay (Leslie et al., 1998; Leslie & Kaldy, 2001; Noles et al., 2005; Pylyshyn, 1989) (Figure 1A).

The way we perceive the world can be attributed to three core cognitive capabilities, these are: (1) our ability to group visual stimuli into objects (Spelke & Kinzler, 2007), (2) our notion of object persistence; from only 2.5 months old, we know that an object that has disappeared from our view continues to exist (Baillargeon, 2002) and (3) our ability to identify multiple instances of the same object over time; this capability is often attributed to "sticky indices" (Leslie et al., 1998; Leslie & Kaldy, 2001; Noles et al., 2005; Pylyshyn, 1989). These indices may be thought of as pointers that, once assigned to an object, remain attached to it, thus uniquely identifying objects over time and space.

Recently, in the field of deep learning, notable progress has been made towards the first of these three core cognitive abilities; several models have been proposed for extracting multiple object representations (Engelcke et al., 2019; Greff et al., 2016) from a static visual scene, these include MONet (Burgess et al., 2019) and IODINE (Greff et al., 2019). These models present compelling unsupervised segmentation results. However, while MONet or IODINE can be used to extract objects from an agent's visual input over time, neither model determines the correspondence between one time step and the next. For example, if MONet or IODINE were used to extract entities from the frames shown in Figure 1A, the model would not know that the purple cone at $t = 0$ was the same as the purple cone at $t = 1$.

In this paper, we take steps towards developing an agent with the second two core cognitive abilities mentioned above: object persistence and "sticky indices". To this end we propose a model with three key properties: (1) a persistent slot-wise object-based memory that stores unique objects in

unique slots, corresponding to their index, (2) the ability to assign a new index (or slot in memory) to a new object and (3) the ability to recognise when an entity is an instance of an object that has been seen before, assigning it the same index (or slot in memory) that was assigned to that object when it first appeared (see Figure 3A). Our model, the AlignNet, and accompanying object-based memory exhibit these properties.

More concretely, the AlignNet is a model for dealing with alignment in the cases where two sets being aligned may contain varying numbers of objects and each set may contain objects that are not present in the other. Additionally, when traditionally solving the alignment problem[1] (e.g. with the Hungarian algorithm), an adjacency (or similarity) matrix is provided. Construction of this adjacency matrix requires hand-crafting a similarity measure. The AlignNet bypasses the need for hand-crafting a similarity measure, by implicitly learning a similarity measure in order to solve the alignment problem. Finally, once trained, the AlignNet may be used to write single entities to single slots in memory, inducing a persistent object-based memory.

Our contributions are as follows:

1. We are the first to formulate the alignment problem such that a neural network may learn a similarity measure for aligning entities (Section 3). We propose the self-supervised Align-Net, a model capable of finding entities in one set that correspond with entities in another set, while also being able to deal with the adding and dropping of entities (Section 3).

2. We demonstrate that a trained AlignNet may be used to write newly appearing objects to empty slots (or indices) in memory and keep track of objects over time (Section 3.2 & 4.4).

3. We show that the AlignNet implicitly learns a similarity measure for aligning entities, tolerant to noise, that outperforms hand-crafted similarity measures (Section 4.1 Figures 6 & 9).

4. We also show that, by using multiple message passing steps, our model learns to attend sharply to single object slots (Section 4.3 and Figure 7c), while remaining differentiable.

5. Our experiments, on a partially observable question-answering dataset, demonstrate the benefits of having a persistent object-based memory (Section 4.5).

We begin by introducing the general alignment problem.

## 2 THE GENERAL ALIGNMENT PROBLEM

Given two sets of entities, $\mathcal{U} = \{u_0, ..., u_{N-1}\}$ and $\mathcal{S} = \{s_0, ..., s_{N-1}\}$, which contain representations for instances of the same $N$ unique objects, the alignment problem is equivalent to finding the *aligning indices*, $I = [i_0, ..., i_{N-1}]$, such that, $u_j \approx s_{i_j}$ for $j = 0, 1, ..., N-1$[2]. The representations are only approximately equal because the object representations may be noisy, meaning that the representation for two similar objects may differ.

While in the case above, $\mathcal{S}$ and $\mathcal{U}$ contain similar entities, they may also be thought of as sets of objects observed in a scene, by an agent, at different points in time. As an agent moves within its environment we expect the number of objects within its field of view to change from one time step to the next (Figure 1A.) which means that if we are aligning sets of objects extracted at different points in time, each set is likely to have different numbers of objects. This means that the AlignNet, and accompanying object-based memory, needs to be able to account for:

- Objects that disappear from one time step to the next, such as an object leaving the field of view after a head movement. We refer to objects that disappear as *dropped* objects.

- New objects that appear from one time step to the next. We refer to new objects as *added* objects.

- Objects that reappear after being hidden for a number of time steps, which requires memory.

---

[1]The Alignment problem may also be referred to as the assignment problem.

[2]In Numpy, Tensorflow and PyTorch this would be this would be implemented as `S[I]`, `tf.gather(S, I)` and `torch.gather(S, I)`, respectively.

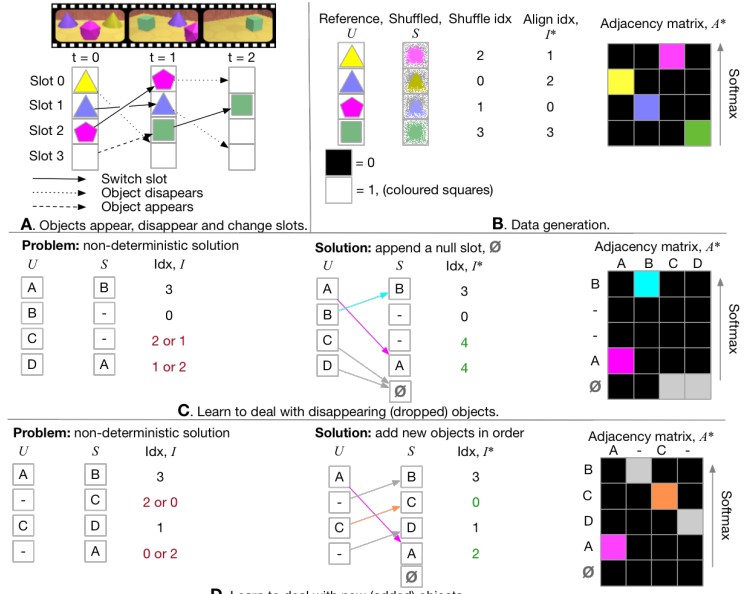

Figure 1: **Our solution to the alignment problem. A**. A schematic demonstrating objects appearing and disappearing over time. **B**. An example of how an a shuffled list, $S$ and aligning indices, $I$ are constructed from an *unshuffled* list $U$, also in the case where objects are **C**. dropped and **D**. added. Note that each of the four columns of $A^*$ is a one-hot vector corresponding to the four indices in $I^*$.

In this paper, we propose a model that, given two sets of object representations, learns to predict the *aligning indices* in the general case, where objects may have been added (Section 3.1.2) or dropped 3.1.1). We then propose to use the AlignNet to align sets of objects over time, using an object-based memory (Section 3.2).

# 3 THE SELF-SUPERVISED ALIGNMENT MODULE, ALIGNNET.

## 3.1 LEARNING TO ALIGN TWO SETS OF OBJECTS VIA SELF-SUPERVISION

We begin with the simple case, where the two sets $\mathcal{U}$ and $\mathcal{S}$ contain the same objects. We use the term object to refer to a vector representing an object. A set of objects, $\mathcal{U}$, may be converted to an *unshuffled* list, $U$, by concatenating[3] the elements in an arbitrary order, $U = [u_0, ..., u_{N-1}]$. To obtain a list of *shuffled* objects, $S = [s_0, ..., s_{N-1}]$, we can shuffle elements in $U$ and add noise. Formally, to obtain $S$ we sample a random permutation matrix, $A^*$, and some noise $\epsilon$ to obtain $S = UA^{*\top} + \epsilon$. The true alignment indices, $I^*$, are therefore given by $I^* = \arg\max A^*$. The argmax is taken over the columns. For an example, see Figure 1B.

It may be tempting to find an adjacency matrix, $A$, that minimises the difference between $SA$ and $U$. However, firstly, there is no obvious way to quantify this similarity and secondly, it is often the case that an $A$ that minimises this difference is not a permutation matrix. Finally, depending on how the similarity is defined, many different solutions may be found that minimise the difference, when in fact there is only one correct (alignment) solution. Rather, our goal is to learn a model, $f_\theta$, such that $A = f_\theta(U, S)$, where each column of $A$ is a categorical distribution over the alignment indices for each element in $S$, by maximising the expectation, $\mathbb{E}_{A^*} \log[f_\theta(U, S)]$.

We choose to model $f_\theta$ as a graph network applied to a bipartite graph[4] (Battaglia et al., 2018). Graph networks take three inputs: nodes, edges and a global variable. We represent the nodes as the concatenation of shuffled and unshuffled objects, $nodes = [S, U]$, where every node in $\mathcal{U}$ is connected to every node in $\mathcal{S}$ by two edges, one going in each direction. There are no connections

---

[3] $[\cdot]$ is the concatenation operation.
[4] For details of graph network computation please refer to Battaglia et al. (2018)

between nodes within $\mathcal{U}$ or $\mathcal{S}$. We do not use the global variable. In our case $|\mathcal{U}| = N$ and $|\mathcal{S}| = N$ and the output edges, $E_{output} \in \Re^{2N \times 2N}$. We take the upper-right quarter of this matrix as the predicted adjacency matrix, $A \in \Re^{N \times N}$ and apply a softmax across the columns of $A$. Recall that the predicted alignment indices are given by, $I = \arg\max A$.

### 3.1.1 LEARNING TO ACCOUNT FOR DISAPPEARING (DROPPED) OBJECTS

Here we extend the model to account for objects that disappear between two time steps; we think of each position in a list as a slot that can either contain an object or be empty. To learn to account for disappearing (dropped) objects during self-supervised training, we can construct an $S$ that is not only a shuffled, corrupted version of $U$ but also has some elements that are *dropped*. Consider an example $U = [A, B, C, D]$ and $S = [B, -, -, A]$, where " $-$ " represents an empty slot. In this case the aligning indices, $I^*$, are ambiguous. The first two aligning indices are straightforward: A has moved to slot three and B has moved to slot zero, leaving slots one and two empty. We could assign C and D to either (or both) of the empty slots. This leads to ambiguity in the solution, creating a problem illustrated in Figure 1C.

To avoid ambiguity in the solution when objects are dropped, we propose to append an additional null slot to $S$ (see the Solution in Figure 1C.), so that all dropped objects can be indexed by this one null slot, making the solution deterministic. The index of the null slot is $N$, the number of objects and therefore the aligning indices for the example above are $[3, 0, 4, 4]$, since $N = 4$.

In this updated formulation, the input nodes to the graph network, $f_\theta$, are now $nodes = [S, U, \varnothing]$, where $\varnothing$ is the additional null slot, and the input and output edges are now $E_{input}, E_{output} \in \Re^{(2N+1) \times (2N+1)}$. We take the upper-right quarter of the matrix as a prediction for $A \in \Re^{(N+1) \times N}$. The adjacency matrix for the problem above is illustrated in Figure 1C. Note that while the columns still sum to one, the rows do not, because the *null slot*, $\varnothing$, may be aligned with slots in $U$. This may be formalised as follows: (1) $\sum_{i=0:N} A_{i,j} = 1$ for $j = 0, 1, ..., N - 1$, this ensures that the columns of $A$ are distributions over the shuffled slots, (2) $0 \leq \left( \sum_{j=0:N-1} A_{i,j} \right) \leq 1$ for $i = 0, 1, ..., N - 1$, this term allows for the special case where if slot $s_i$ is empty, the sum across row $i$ should be zero, and (3) $\sum_{j=0:N-1} A_{N,j} = n_{dropped}$, for $n_{dropped} \leq N$, this final term, says that if $n_{dropped}$ objects have been dropped from $U$, then the sum over the $N^{th}$ row will be $n_{dropped}$.

### 3.1.2 LEARNING TO ACCOUNT FOR NEWLY APPEARING (ADDED) OBJECTS

Now we consider how to deal with newly appearing objects. To learn to account for appearing (added) objects during self-supervised training, we can construct a $U$ that is missing certain elements that are present in $S$. We consider these to be the *added* elements. Consider an example (illustrated in Figure 1D.) $U = [A, -, C, -]$ and $S = [B, C, D, A]$. In this case B and D are the added objects. Again, accounting for A and C is easy; A has moved to slot 3 and C has moved to slot 1. However, B and D could be assigned to either slot 0 or slot 2. Additionally, a single object may be assigned to both slots. It is imperative that this does not happen, but is hard to enforce through regularisation alone.

Our solution, when adding objects, is to remove ambiguity by training the model to assign the first new object that it encounters in $S$ to the first available empty slot in $U$. By imposing an ordering, we ensure that there is only one solution. For the model to be able to infer the order of the slots, we append slot indices to the object representations in $S$ and $U$ (after shuffling).

Finally, in order to add a new object to a new slot, is it necessary to first determine whether the object *is* new. To infer if an object is new, we hypothesise that it is not sufficient to consider first-order similarity between objects. Knowing that $s_0$ is not similar to $u_i, \forall i$, is not sufficient to say that, $s_0$ is a new object. To be confident that $s_0$ is new object, we have to be sure that it is less similar to $u_i, \forall i$ than any other element in $S$. This reasoning requires the model to understand *relative similarity*, not just the similarity between independent pairs of entities. Therefore, we expect that a single step of graph network computations will not be sufficient to solve the alignment problem. Rather, we expect that recurrent application of a graph network, often referred to as *message passing*, will be needed to infer information about relative similarity.

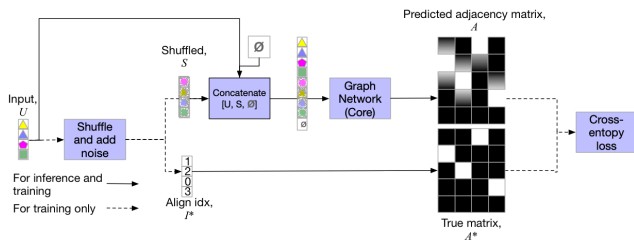

Figure 2: **Schematic of the self-supervised AlignNet.**

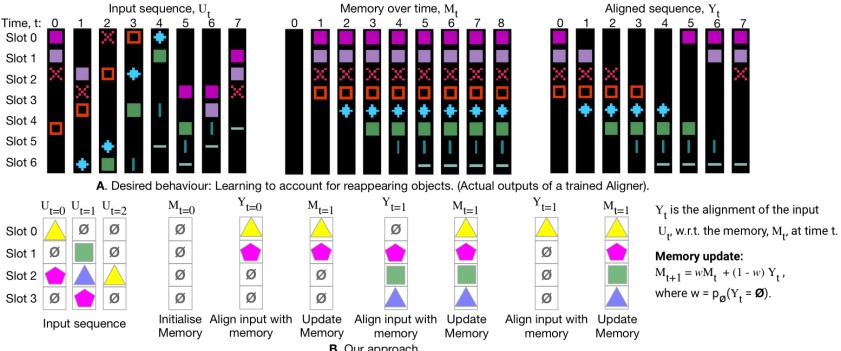

Figure 3: **The Goal: Aligning a sequence of sets of entities.** A. Given an input sequence of sets of objects (left), we would like to obtain a sequence of objects that are aligned over time (right). This means that once a new object appears it is assigned a "sticky index" (Pylyshyn, 1989), corresponding to a slot index, any subsequent instances of that same object must be assigned to that same slot. B. To achieve this we propose a two-step process: (1) Align inputs with respect to the memory, (2) update the memory with the new objects.

In summary, our model, the AlignNet (Figure 2), is able to deal with dropped objects, by appending a null slot, $\varnothing$ to $S$, allowing entities that have been dropped to be assigned to this slot. Our model is also able to deal with objects that have been added, by imposing an order in which new objects are assigned to empty slots.

## 3.2 LEARNING TO ALIGN SEQUENCES OF SETS OF ENTITIES

The model presented so far does not account for objects that disappear and reappear at some point in the future, so here we describe how to incorporate that functionality. Given a sequence of sets of objects, $[\mathcal{U}_{t=0}, ..., \mathcal{U}_{t=T-1}]$, where $T$ is the number of time steps, we would like to assign each unique object in $\cup_{\tau=0:T-1}\mathcal{U}_{t=\tau}$ a slot index, which the object maintains over time, $t$. This is illustrated in Figure 3A. The slot indices are analogous to the "sticky indices" referred to by Leslie et al. (1998) and Pylyshyn (1989). To achieve this, the model needs to not only account for the appearance of new objects or the disappearance of objects; the model also has to account for the reappearance of objects. To deal with reappearing objects, the model requires a memory, $M$, to recall which slot (or index) each object belongs to. We propose an iterative two-step process for aligning sequences of sets of entities over time. In the first step, the input is aligned with the memory, $M$, using a trained AlignNet. In the second step the memory is updated with the new objects.

More formally, let $S_t$ be the input and $M_t$ be the memory at the time, $t$, where $M_0$ has $N$ empty slots. At each time step, we compute an adjacency matrix, $A_t = f_\theta(M_t, S_t)$ and obtain a list of aligned objects, $Y_t$, using hard alignment[5]. The memory is updated, $M_{t+1} = wM_t + (1 - w)Y_t$ where $w$ is the last row of the predicted adjacency matrix and represents the probability, $p_\varnothing(Y_t = \varnothing)$, that a slot is empty. Note that the parameters, $\theta$, of the trained AlignNet, $f_\theta$, are held fixed and there is no additional training, these steps are illustrated in Figure 3B.

---

[5]Hard alignment: `tf.gather(`$S_t$`, argmax(`$A_t$`))`.

Table 1: **Comparing the AlignNet to hand-crafted baselines.** Each experiment was run five times and the mean is shown here. (Training curves with standard deviations across runs are show in Figure 7, under $max\_num\_to\_drop = 0$)

|  | Accuracy | Entropy | Difference between hard and soft alignment MSE |
|---|---|---|---|
| AlignNet (Ours) | **99.9%** | **0.0** | **0.0** |
| MSE | 99.2% | 14.6 | 1.8 |
| Cosine | 96.5% | 15.8 | 2.6 |

In sections 3.1.1 and 3.1.2 we only considered cases where objects are dropped from – or added to – a slot. However, it is often the case that some slots in $M_t$ are empty and remain empty. Therefore when training an AlignNet it is necessary to include cases where a slot remains empty from one time step to the next. For example, $U = [A, B, -, D]$ and $S = [A, D, B, -]$, the third slot in $U$ is empty and there is no new object in $S$ that could be assigned to the third slot.

## 4 EXPERIMENTS AND RESULTS

We test our model in various different settings, isolating different behavioural properties and compare to baselines. We separately show that the AlignNet is able to account for both dropped (Section 4.2) and added (Section 4.3) objects. We then demonstrate how a trained AlignNet trained with both added and dropped objects may be used to align sequences of sets of objects (Section 4.4). In our experiments, we report the alignment accuracy and the entropy over the columns of the predicted adjacency matrices.

In the experiments that follow we use a symbolic dataset that represents objects in a partially observable room that contains eight objects. A random agent moves in the room collecting observations. Each observation is a set of entities, $\mathcal{U}$, where each $u_j$ is a six element vector representing the colour (three elements), shape and position (two elements) of the objects. The elements in the vectors are normalised between $[0, 1]$. We add uniform noise to each object representation. The level of noise varies depending on the experiment. Details of the AlignNet architecture can be found in the appendix G.

### 4.1 BASELINES AND RESULTS WHEN NO OBJECTS ARE ADDED OR DROPPED

When traditionally solving the alignment problem, a similarity matrix, also referred to as an adjacency matrix, is provided. Construction of this matrix requires hand-crafting a similarity measure. The AlignNet bypasses the need for this by implicitly learning a similarity measure suitable for solving the alignment problem.

We compare the AlignNet to a model that uses hand-crafted similarity measures to construct the adjacency matrix; the cosine and mean-squared-error, MSE, distances. For the baselines we align two sets that have the same objects (no added or dropped objects) with noise, $Uniform(-0.1, 0.1)$, added to the object representations.

Results in Table 1 show that while the hand-crafted methods achieve competitive accuracy, the entropy of the slot prediction is very high, meaning that differences between hard and soft alignment are large. While hard alignment assigns a single element in U to a single element in S, for our system to remain differentiable, soft alignment is preferable. We demonstrate that our model achieves similar performance for both hard and soft alignment[6].

The baseline models are not designed to deal with cases where there are different numbers of elements in each set, $\mathcal{U}$ and $\mathcal{S}$, or where there is no correspondence for some elements. The AlignNet is able to cope with both of these cases. In the next section, we demonstrate the AlignNet's unique ability to deal with cases where elements in set $\mathcal{U}$ are not present in set $\mathcal{S}$, we refer to this as *dropping* (dropping was explained in Section 3.1.1).

---

[6]Hard alignment: `tf.gather(S, argmax(A))`, Soft alignment: `tf.matmul(S, A)`.

Table 2: **Comparing the AlignNet to hand-crafted baselines for adding and dropping.** The AlignNet achieves higher accuracy when objects are added and dropped compared to baselines. (See Figures 7 and 8 in the Appendix for more extensive results with more objects added and dropped.)

| % accuracy | $max\_num\_to\_add =$ | 0 | 0 | 1 | 2 |
|---|---|---|---|---|---|
| | $max\_num\_to\_drop =$ | 0 | 1 | 0 | 2 |
| AlignNet (Ours) | | **99.9%** | **99.9%** | **99.9%** | **99.9%** |
| MSE | | 99.2% | 91.2% | 93.7% | 88.5% |
| Cosine | | 96.5% | 87.9% | 91.1% | 86.2% |

## 4.2 RESULTS FOR DISAPPEARING (DROPPED) OBJECTS

We train separate AlignNets to drop up to five objects (i.e. $max\_num\_to\_drop \in \{1, 2, 3, 4, 5\}$), where at most $max\_num\_to\_drop$ objects that were present in set $\mathcal{U}$ are replaced with empty slots in set $\mathcal{S}$. A small amount of uniform noise, $\epsilon \sim Uniform(-0.1, 0.1)$, was added to each object representation in each set. The AlignNet achieves 99.9% alignment accuracy when dropping up to five objects and the differences between hard and soft alignment are negligible (Figure 7 in the Appendix). We use the same hand-crafted baselines as before to give our results context. Our model significantly outperforms the baselines which are not able to handle cases where objects have been dropped (see Table 2 for a brief summary).

Additionally, in Figure 6 we show the effect on the AlignNet's performance – when trained with at most two objects being dropped – with different amounts of noise $\epsilon \sim Uniform(-noise\_level, noise\_level)$ added to the object representations. We see that, the Align-Net learns to be robust to $0.01 \leq noise\_level \leq 0.2$, while the baselines perform poorly. This is because the AlignNet learns its own similarity function, invariant to noise, rather than using a hand-crafted similarity function that is not invariant to noise.

Finally, we consider how well an AlignNet trained with $max\_num\_to\_drop = 2$ generalises to $2 < max\_num\_to\_drop \leq 7$. We found that an AlignNet trained on $max\_num\_to\_drop = 2$ generalises with almost perfect accuracy when tested with $2 < max\_num\_to\_drop \leq 7$ (See Table 3). This may be attributed, in part, to the graph network, which shares parameters across nodes and edges, applying the same model to all nodes, independent of their order.

Now we move on, demonstrating the AlignNet's unique ability to cope with new objects appearing in set $\mathcal{S}$ (that are not present in set $\mathcal{U}$). The AlignNet must learn to assign new objects in $\mathcal{S}$ indices that correspond to empty slots in $\mathcal{U}$ (adding new objects was described in Section 3.1.2).

## 4.3 RESULTS FOR NEW OBJECTS APPEARING (ADDED)

For each $max\_num\_to\_add \in \{1, 2, 3, 4, 5\}$ we trained five AlignNets, for each $max\_num\_to\_add$ at least one of the five models achieves an accuracy $> 99\%$ (see Figure 8). Figure 9 demonstrates that the AlignNet is able to learn a similarity measure that is tolerant to noise; the AlignNet is able to achieve more than 99% accuracy when trained with different levels of noise.

One interesting finding was that, in order to achieve maximal accuracy and low entropy solutions, multiple message passing steps, in the form of recurrently applying the graph network to the input nodes, were needed. Figure 12(b) in the Appendix shows that at least two message passing steps ($num\_reccurent\_steps = 2$) are needed to achieve 99.9% accuracy. Multiple message passing steps allow to the model to learn higher order similarities between entities. The model does not *just* learn that two entities are similar, but that entities are more or less similar than other entities. We found that two or more message passing steps also led to very low entropy (sharp attention) solutions, resulting in hard and soft alignment being indistinguishable from each other. Again, this would be useful if the AlignNet were to be incorporated into a larger architecture since soft alignment is differentiable.

We also consider how well a model trained on examples where at most $\mu$ objects were added ($max\_num\_to\_add = \mu$), generalises to cases where more than $\mu$ objects are added. The results are shown in Table 4 of the Appendix. We see that AlignNets trained with larger $\mu$ generalise better.

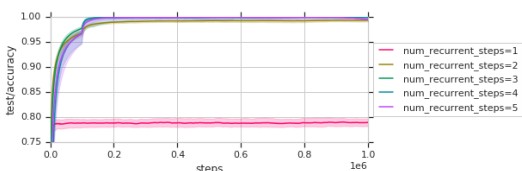

Figure 4: **How many message passing steps are needed?** For each $num\_recurrent\_steps$ we trained five AlignNets and show the median performance with the standard deviations. The Aligner is trained with random numbers of objects being added, dropped and slots left empty. The top five models all achieve 99.9% accuracy.

We have shown that the AlignNet can learn to add and drop a variable number of objects and is also tolerant to noise. In the next section we look at training an AlignNet, which we will refer to as the *Aligner*. We then use the Aligner to write to an external slot-based memory, without any further training. We begin by training the Aligner.

### 4.4 TRAINING ALIGNERS FOR WRITING TO MEMORY AND RESULTS ALIGNING SEQUENCES OF SETS OF ENTITIES

When using a trained Aligner to write to memory, $M_t$, the Aligner needs to be able to align an input, $S_t$, with the memory. The Aligner must be able to deal with adding and dropping as well as dealing with empty slots. The Aligner is trained, as before, to align shuffled lists, $S$, with unshuffled lists, $U$. We must therefore ensure that pairs of $S$ and $U$ exhibit added objects, dropped objects and empty slots (empty slots were explained in Section 3.2).

We train our Aligner to add, drop and keep empty a random number of entities and slots. We experimented with different numbers of recurrent, message passing steps (Figure 4) and found that the best models had three or four message passing steps, achieving 99.9% accuracy. Now we explore how well our trained Aligner writes to memory.

Taking the best trained Aligner, we use the model to align sequences of up to seven steps with varying numbers of *kept* objects. The number of kept objects, $num\_keep\_slots$, refers to the number of occupied slots in $S_t$ (Figure 3A. shows an example of $num\_keep\_slots = 4$). The trained Aligner writes to memory with 100% accuracy across seven time steps and for $num\_keep\_slots = \{1, 2, 3, 4, 5, 6, 7\}$, correctly adding all new elements to memory at each time step (see Figures 14 and 15 in the Appendix).

The AlignNet with a slot-wise object-based memory is an example of a persistent object memory. In the next section we demonstrate how a persistent object memory can be useful for solving tasks in partially observable environments that contain objects.

### 4.5 EPISODIC PARTIALLY OBSERVABLE QUESTION ANSWERING

Finally, we explore question answering using the AlignNet for a domain containing objects that appear, disappear and reappear randomly within an episode (Figure 5). We use a dataset called *epsiodic-sort-of-clevr* (see the Appendix H for further details) where the goal is to answer a question about the relationship between objects within an episode. To solve this task, a model must learn to represent objects in memory and then use this memory to answer a question.

The AlignNet together with a relation network (Santoro et al., 2017), AlignNet-RN, can learn to answer questions from episodic-sort-of-clevr, reaching an accuracy of 93% for episodes containing six objects per scene, two scenes per episode, 100 observations per scene and two objects per observation. The AlignNet can accurately align objects across multiple scenes within an episode. Objects that disappear and reappear within a scene are successfully aligned with the memory and thus only new entities are written to empty slots in a fixed sized memory. We compare this to a relation network, RN, without the AlignNet, where all observed objects are added to the memory at each time step (Pritzel et al., 2017). The computational complexity of the RN scales quadratically with the number of input objects, so the computational cost of an RN without an AlignNet is much greater than the AlignNet-RN model. In this example, the AlignNet memory contains 12 object representations (six objects $\times$ two scenes), so the AlignNet-RN must compute $12^2 = 144$ relations. In comparison, the RN without the AlignNet has 400 object representations in memory (100 observations per scene $\times$ two scenes $\times$ two objects per observation). Therefore the RN has to compute $400^2 = 160,000$ relations which becomes increasingly infeasible.

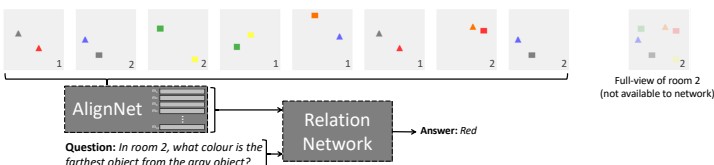

Figure 5: **Episodic-sort-of-clevr question answering with the AlignNet and Relation Network**
The episodic-sort-of-clevr question answering dataset consists of a sequence of observations of objects within a scene, with objects appearing, disappearing and reappearing again across multiple scenes. Each object is represented symbolically and each scene is given a unique label. In this example, the episode contains two scenes, and each scene contains six objects, with four observations per scene and two objects per observation. The AlignNet learns to write objects to an object-aligned memory. This memory is then provided as input to a relation network, along with the question. The relation network is trained to answer questions about the episode.

## 5 RELATED WORK

**Human object individuation and identification.** The AlignNet is inspired by work by Pylyshyn (1989) on "sticky indices", a paradigm for explaining how humans keep track of entities as they go in and out of their visual field. When a new object appears, it is assigned an index. The index then "sticks" to the object, so when the object moves, it retains the same index. The exact mechanism by which the indices "stick" is not clear. Noles et al. (2005) suggest that new object states are compared to old object states and if they are *similar enough* the old states are updated and if they are not similar enough a new index is created. This is similar to the function that the AlignNet learns to perform, using multiple message passing steps to learn when something is, or is not, *similar enough*. Additionally, evidence suggests that we rely on both object properties and their locations to successfully individuate and identify objects (Leslie et al., 1998; Xu, 1999), which is why we use representation of both object properties and locations in our experiments.

**Combinatorics: traditional and deep approaches.** The AlignNet is unique, compared to other traditional and deep learning approaches for solving combinatorics problems because it does not assume access to a similarity or adjacency matrix (Bello et al., 2016; Vinyals et al., 2015; Milan et al., 2017), which is required for algorithms such as the Hungarian algorithm. The AlignNet simultaneously learns an implicit similarity measure that helps it to solve the assignment problem. Additionally, we consider a more general assignment problem where sets may have additional elements for which there is no match.

Interesting work by Lyu et al. (2019) demonstrates that deep networks can learn permutation matrices that, like the AlignNet, have low entropy solutions. However, their work does not focus on alignment and the adjacency matrices in the AlignNet are not permutation matrices because an additional row is appended to deal with *dropped* objects. Andrychowicz & Kurach (2016) also develop an interesting model for sorting and merging sequences. However their proposed method is non-differentiable and does not focus on alignment.

**Object tracking: deep learning techniques.** A key novel feature of the AlignNet is its ability to deal with both adding (appearing) and dropping (disappearing) objects. He et al. (2018) propose a model for tracking objects using a memory addressing mechanism. The model terminates trackers when an object disappears, therefore the model cannot deal with reappearing objects, while the AlignNet can. Additionally, they assume that all the entities visible at $t = 0$ are all the objects visible across time, meaning that the model cannot deal with new objects appearing. In the object tracking literature, when a new object appears, which is not sufficiently similar to the objects that have been seen before, it may be considered to be false a detection (Yoon et al., 2019).

Valmadre et al. (2017); Yang & Chan (2018) also include an external memory for tracking single objects. Their memory is used for evolving templates for matching over time, while the AlignNet memory keeps track of all the entities that have been seen allowing the model to track multiple pre-extracted entities. Our work is quite different to traditional object tracking, in that we focus on tracking pre-extracted entities, while most previous work focuses on tracking from pixels.

**A move towards entities driving a need for alignment.** Several recent works in relational reasoning (Yi et al., 2018; Janner et al., 2018; Ferreira et al., 2019; Co-Reyes et al., 2019) and reinforcement learning (Zambaldi et al., 2018; Kulkarni et al., 2019) have demonstrated the benefit of working at the entity, rather than pixel level. In these applications, the environment is fully observable; objects do not appear or disappear from view.

While models exist for extracting entities (Burgess et al., 2019; Greff et al., 2019; Nash et al., 2017; Greff et al., 2016), as far as we know, there are no models suitable for keeping track of correspondence between entities over time in partially observable environments; this is the purpose of the AlignNet.

**Learning permutation invariant functions on sets and graphs.** At the core of our model is a graph network (Battaglia et al., 2018; Xu et al., 2018) and in the Appendix we also show that a mutli-layer attention network (or transformer) (Vaswani et al., 2017; Wang et al., 2018) works well too (Figure 13). We chose these models because they learn functions that are permutation invariant. This is essential when dealing with sets, which have no order. Many other models also learn permutation invariant functions such as relation networks (Santoro et al., 2017; Murphy et al., 2018; Battaglia et al., 2016), deep sets (Zaheer et al., 2017) and graph attention networks (Veličković et al., 2017). Any of these models may have been suitable to place at the core of the AlignNet. However, we found that the transformers scaled better and the graph networks were easy to train.

Finally, more similar to our own work, Zhang et al. (2019) propose a model for predicting sets. However when determining correspondences between elements in the sets they still use hand-crafted set losses and their model is not a simple feed forward model because they apply gradient descent on the set itself.

**Reading from and writing to memory.** The AlignNet keeps track of multiple pre-extracted entities, over time, with a persistent slot-base memory, where each slot contains the representations for a single object. When writing to memory, new observations should be placed in new slots and observations similar to those already in memory should either be updated or left as they are. There are few examples of models that learn to write to memory (Graves et al., 2014). This is often because learning what and where to write can be challenging. In a special case, Parisotto & Salakhutdinov (2017) propose the Neural Map, a 2D spatial memory where memories are stored based on the location of the agent. Often all observations, along with a look up key, are written to (an episodic) memory (Pritzel et al., 2017; Blundell et al., 2016) using a differentiable neural dictionary, DND. In contrast, the AlignNet has a fixed sized memory and learns to only write new entities to empty slots in memory.

To obtain sharp attention, we argue that knowing the similarity between two things is not sufficient, we also need to know how similar a (key, query) pair is relative to other (key, query) pairs. By incorporating multiple message passing steps, we demonstrate that our model may achieve low entropy (sharp) attention over slots in memory. Another method for sharply attending to slots in memory is variational memory addressing (Bornschein et al., 2017). However, Bornschein et al. (2017) focus on reading from memory, rather than adding new entities to memory.

## 6 CONCLUSION

Our AlignNet model is unique in several ways. It is a model for dealing with alignment in the cases where two sets being aligned may have varying numbers of objects and each set may contain objects that are not present in the other. Additionally, the AlignNet bypasses the need for hand-crafting a similarity measure, implicitly learning a similarity measure in order to solve the alignment problem. Finally, we demonstrate the benefits of having a persistent object-based memory when solving a question answering problem in a partially observable environment.

ACKNOWLEDGEMENTS

Acknowledgements

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

## A  RESULTS FOR DROPPING OBJECTS

Figure 6 shows the effect of different levels of corruption on alignment accuracy; the AlignNet is able to achieve more than $99\%$ test accuracy at all noise levels when at most two objects are dropped.

Figure 7 shows test prediction accuracy and entropy for an AlignNet trained to drop up to five objects.

Table 3 demonstrates that an AlignNet trained to drop N objects, can drop more than N objects with almost perfect accuracy.

Table 3: **How well does a model trained with at most two objects being dropped ($max\_num\_to\_drop = 2$) generalise to cases were more than two objects are dropped?** We trained an AlignNet with $max\_num\_to\_drop = 2$ and tested the model on examples where more than two objects were dropped. We see that the model generalises well to more objects being dropped than those seen during training.

| $max\_num\_to\_drop$ | Test accuracy |
|:---:|:---|
| 0 | 100% |
| 1 | 100% |
| 2 | 99.9% |
| 3 | 99.8% |
| 4 | 99.8% |
| 5 | 100% |
| 6 | 99.6% |
| 7 | 100% |

## B  RESULTS FOR ADDING OBJECTS

Figure 8 shows test prediction accuracy and entropy values for an AlignNet trained to add up to five objects.

Figure 9 shows the effect of different levels of corruption on alignment accuracy; the AlignNet is able to achieve more than $99\%$ accuracy at all noise levels when at most two objects are added.

Table 4 demonstrates that an AlignNet trained to add at most $\mu$ objects, can add more than $\mu$ objects with high accuracy, particularly when $\mu \geq 3$.

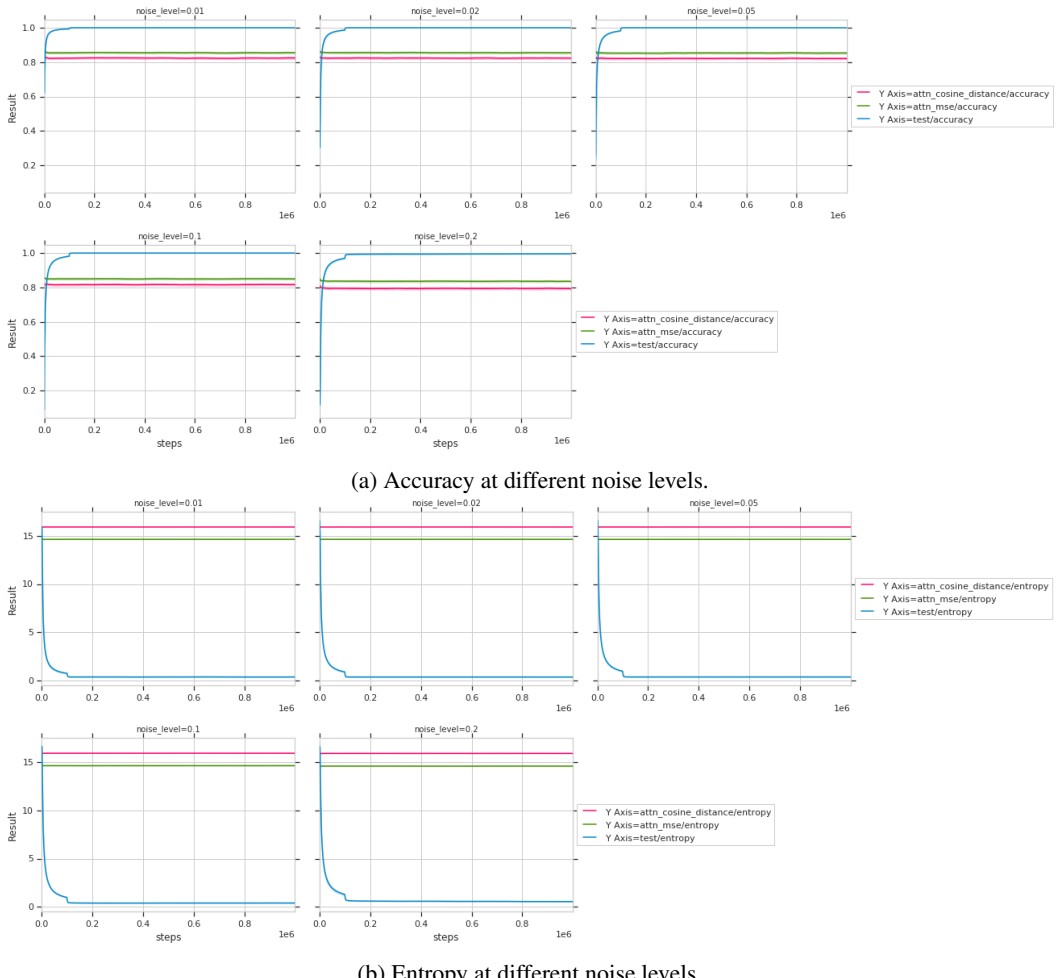

(a) Accuracy at different noise levels.

(b) Entropy at different noise levels.

Figure 6: **Varying the noise level and comparing to baselines for** $max\_num\_to\_drop = 2$**.** For each $noise\_level$ we trained five AlignNets with $max\_num\_to\_drop = 2$ and show the median performance with the standard deviations. The AlignNet achieves $> 99\%$ accuracy at all noise levels and out-performs baselines.

## C    RESULTS FOR BOTH ADDING AND DROPPING OBJECTS

In most real world cases, it is likely that objects are both added to and dropped from $U$ to give an $S$ that has both empty slots and new entities. We explore this case here.

We assess the AlignNet's ability to deal with cases where objects are both added and dropped. We show the effect of different levels of corruption ($noise\_level = \{0.01, 0.02, 0.05, 0.1, 0.2\}$) on alignment accuracy when at most two objects are added and at most two objects are dropped (See Figure 10). Again, we found that two recurrent, message passing steps were necessary and sufficient to achieve maximal performance and low entropy (sharp attention) solutions. The AlignNet achieves more than $99\%$ accuracy at all noise levels when at most two objects are added, except at $noise\_level = 0.2$, where the best model achieves $98.7\%$ accuracy. Visualisations of the results are shown in Figure 11.

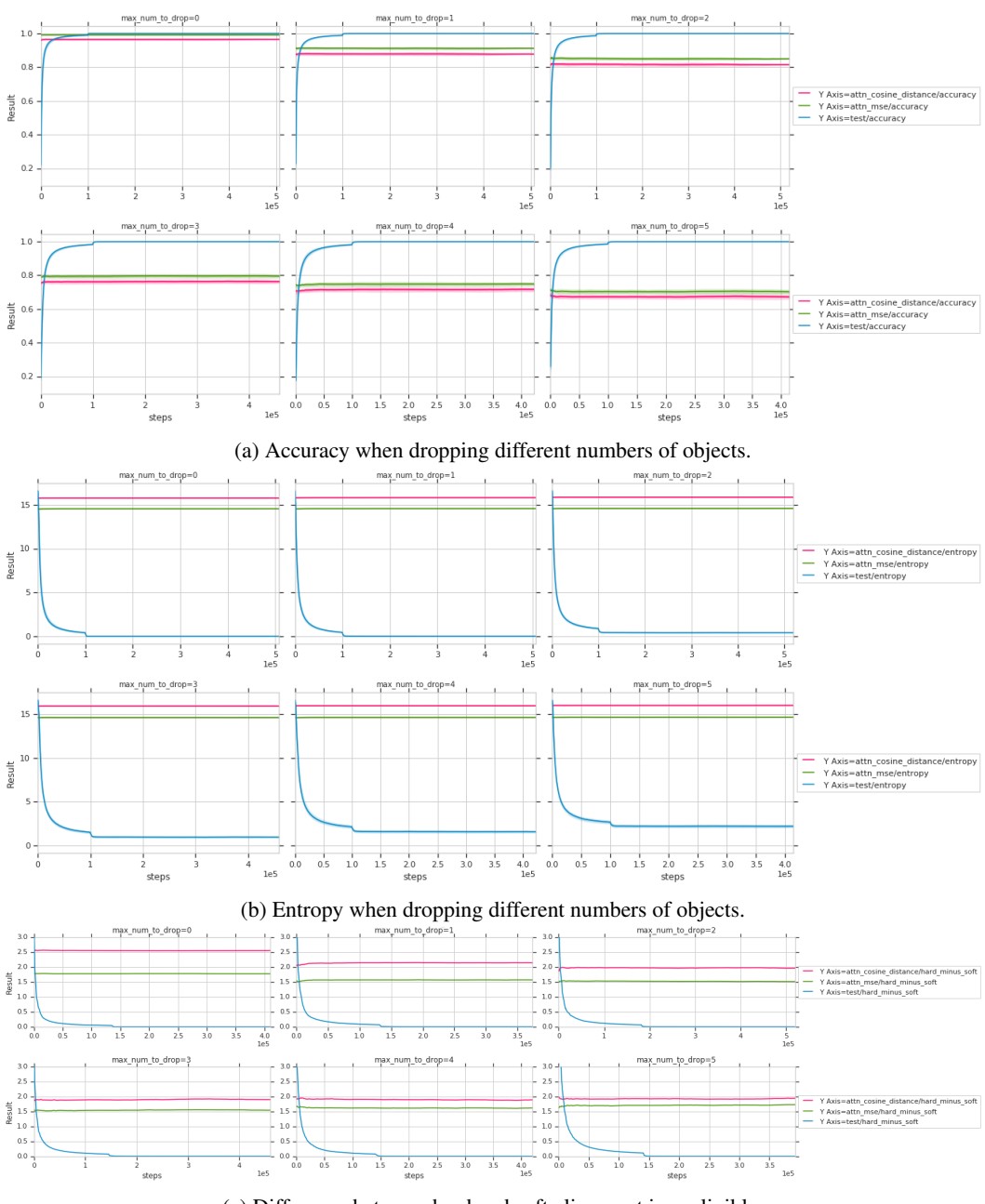

(a) Accuracy when dropping different numbers of objects.

(b) Entropy when dropping different numbers of objects.

(c) Difference between hard and soft alignment is negligible.

Figure 7: **Dropping up to five objects and comparing to baselines.** For each $max\_num\_to\_drop$ we trained five AlignNets and show the median performance with the standard deviations. We also take the median across five runs of the baseline models. The AlignNet achieves $> 99.9\%$ accuracy when dropping up to five objects.

# D  HOW MANY MESSAGE PASSING STEPS ARE NEEDED?

Figure 12 shows that, while only one message passing step is needed when dropping objects, at least two message passing steps are needed when adding objects. Additional message passing steps can make training less stable.

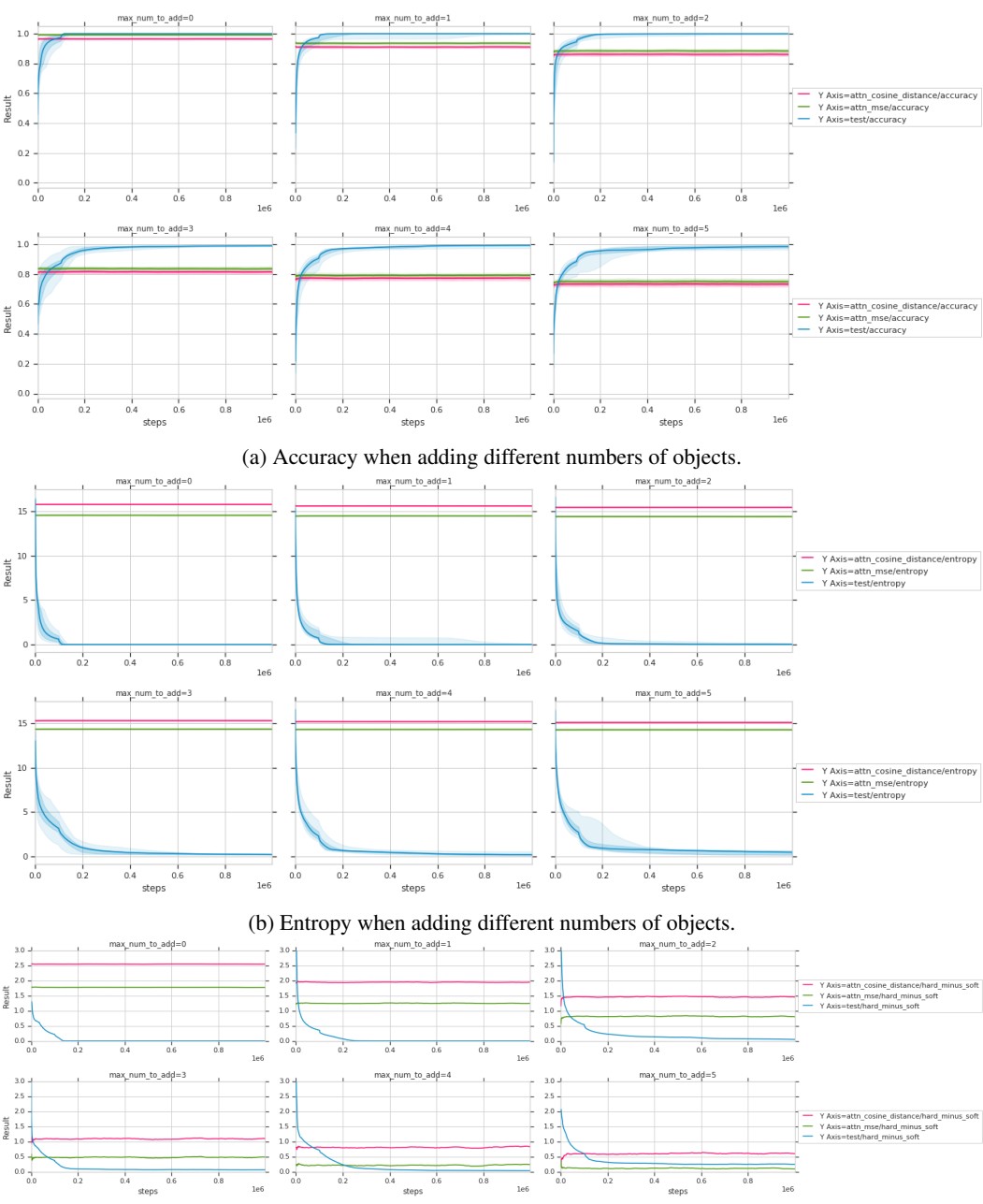

(a) Accuracy when adding different numbers of objects.

(b) Entropy when adding different numbers of objects.

(c) Difference between hard and soft alignment is negligible.

Figure 8: **Adding up to five objects and comparing to baselines.** For each $max\_num\_to\_add$ we trained five AlignNets and show the median performance with the standard deviations. We also take the median across five runs of the baseline models. For each $max\_num\_to\_add$ at least one of the five models achieves an accuracy $> 99\%$.

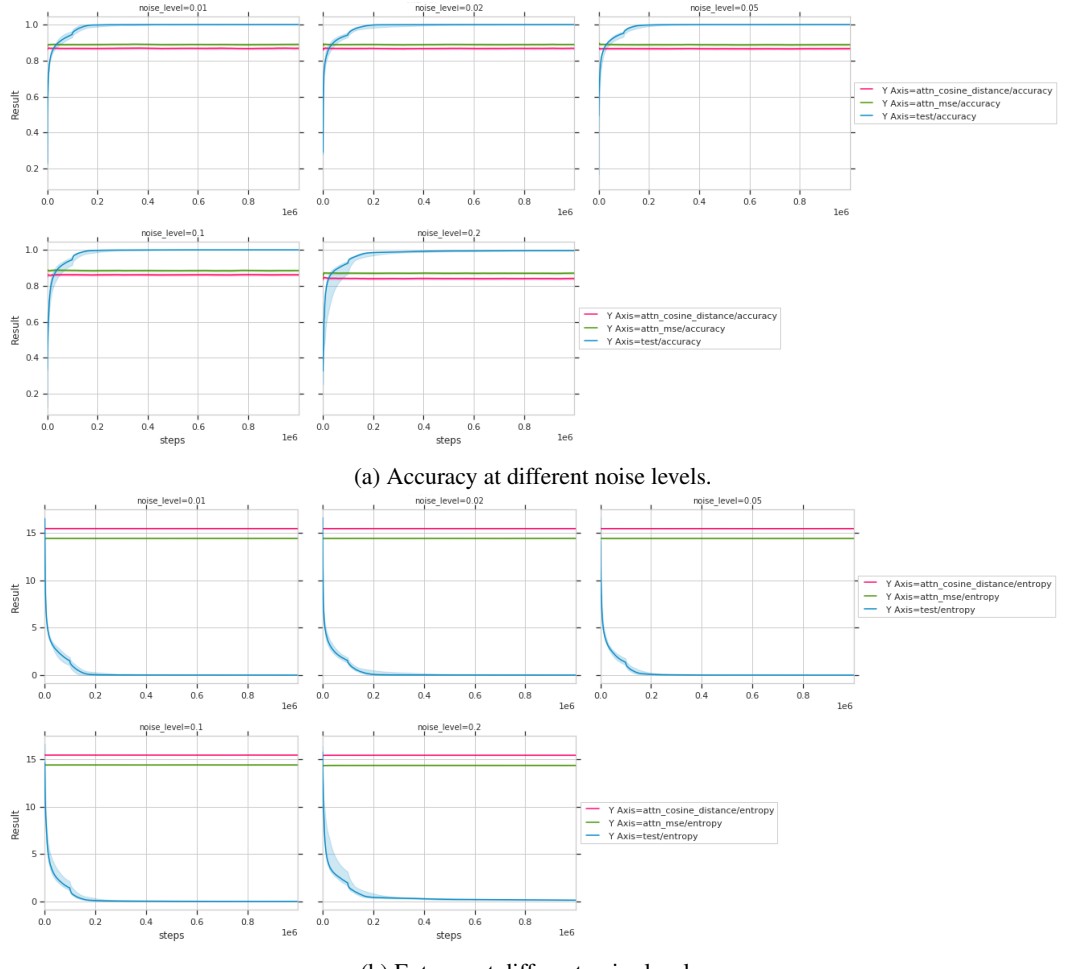

(a) Accuracy at different noise levels.

(b) Entropy at different noise levels.

Figure 9: **Varying the noise level and comparing to baselines for** $max\_num\_to\_add = 2$**.** For each $noise\_level$ we trained five AlignNets with $max\_num\_to\_add = 2$ and show the median performance with the standard deviations. We also take the median across five runs of the baseline models. The AlignNet achieves $> 99\%$ accuracy at all noise levels, out-performing baselines.

Table 4: **How well does an AlignNet trained with at most** $max\_num\_to\_add = \mu$ **objects added, generalise to cases where more than** $\mu$ **objects are added** ($\mu \leq max\_num\_to\_add \leq 7$)**?** Test accuracy values shown in pink are those for which the model is seeing more added objects than those seen during training.

| Maximum number of objects added, $\mu$ | Test accuracy when $\mu$ objects are added using a model trained to add at most | | | |
|---|---|---|---|---|
| | two objects | three objects | four objects | five objects |
| $\mu = 0$ | 100% | 100% | 100% | 100% |
| $\mu = 1$ | 100% | 100% | 100% | 100% |
| $\mu = 2$ | 100% | 100% | 100% | 100% |
| $\mu = 3$ | 86% | 96% | 100% | 100% |
| $\mu = 4$ | 75% | 90% | 99% | 99% |
| $\mu = 5$ | 70% | 80% | 95% | 98% |
| $\mu = 6$ | 61% | 70% | 93% | 98% |
| $\mu = 7$ | 56% | 63% | 90% | 96% |

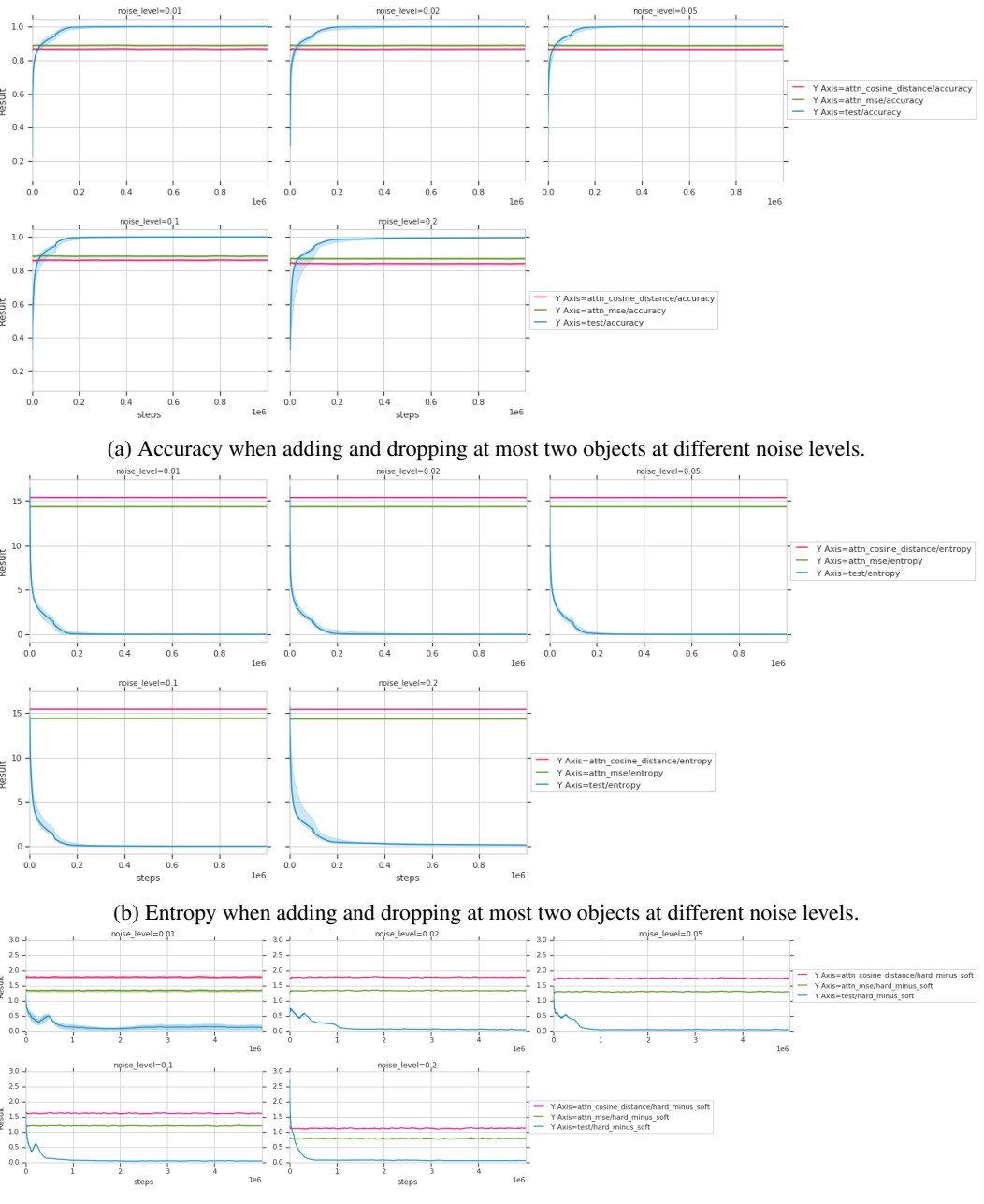

(a) Accuracy when adding and dropping at most two objects at different noise levels.

(b) Entropy when adding and dropping at most two objects at different noise levels.

(c) Difference between hard and soft alignment.

Figure 10: **Varying the noise level and comparing to baselines for** $max\_num\_to\_add = 2$ **and** $max\_num\_to\_drop = 2$. For each $noise\_level$ we trained five AlignNets and show the median performance with the standard deviations. We also take the median across five runs of the baseline models. The best model at each noise level achieves $> 99\%$ accuracy at all noise levels, except at $noise\_level = 0.2$, where the best model achieves $98.7\%$ accuracy. All of our models out-perform baselines.

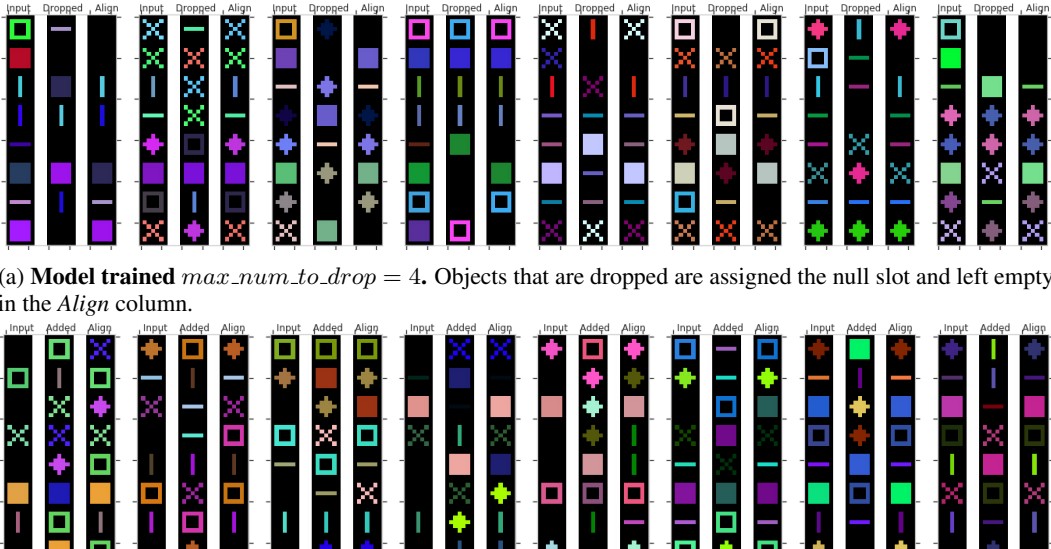

(a) **Model trained** $max\_num\_to\_drop = 4$. Objects that are dropped are assigned the null slot and left empty in the *Align* column.

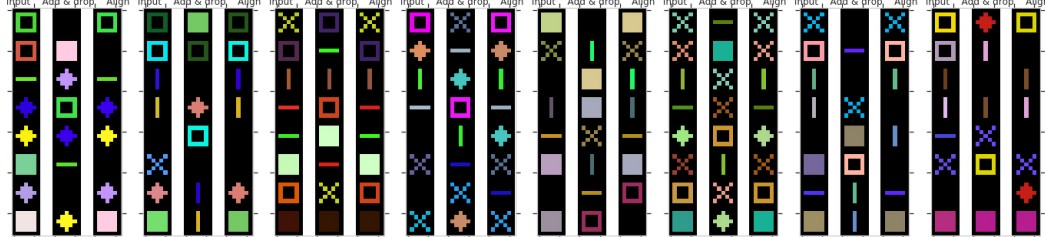

(b) **Model trained with** $max\_num\_to\_add = 4$. New objects are added in the order in which they appear.

(c) **Model trained with** $max\_num\_to\_add = 2$ **and** $max\_num\_to\_drop = 2$.

Figure 11: **Visualisations of AlignNet results.** These examples are not cherry picked. The *Input* column shows the list of objects, $U$, the *Dropped* column shows the list of objects, $S$, with entities dropped and the *Add* column shows the list of objects, $S$ with additional entities not present in the *Input*. The *Align* column shows the aligned entities, slots that were empty in the *Input* may now be occupied by new objects and entities that were in the *Input* but dropped are replaced by empty slots. Entities present in both the *Input* and the *Added* (and) or *Dropped* columns are placed in the same slot that they were found in, in the *Input*.

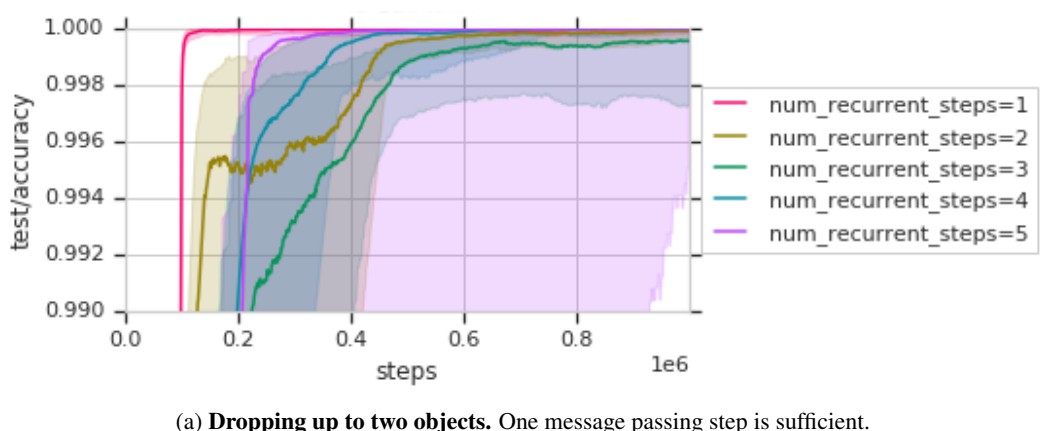

(a) **Dropping up to two objects.** One message passing step is sufficient.

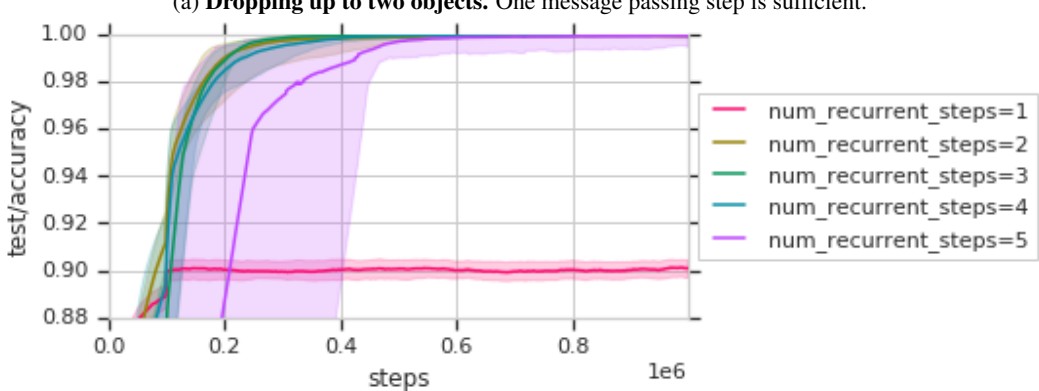

(b) **Adding up to two objects.** At least two message passing steps are needed.

Figure 12: **Comparing the number of message passing steps needed to obtain high test accuracy when adding and dropping objects.** For dropping, a single message passing step is sufficient. However, for adding at least two message passing steps are needed to obtain good performance. More than two message passing steps may results in instability and higher variability during training.

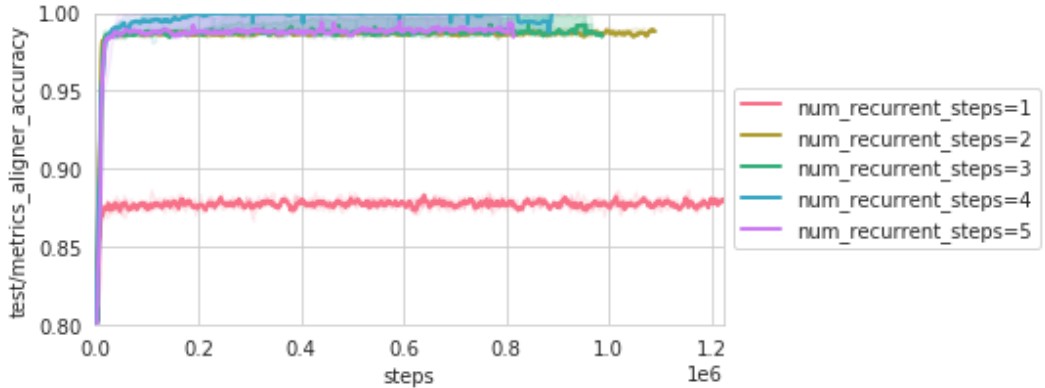

Figure 13: **The AlignNet as a general framework with transformers at the core,** rather than a graph network. We found that at least two layers of transformers were needed to achieve maximal performance.

## E    THE ALIGNNET AS A GENERAL FRAMEWORK

The AlignNet may be more broadly thought of as a framework with the following key elements: (1) transforming sets in to lists of slots, with a fixed order, sorting one list with respect to another, (2) a deterministic order for adding new objects, (3) appending an additional *null* slot for modelling dropped objects and (4) computing a loss between the predicted adjacency matrix and the true adjacency matrix. The model that takes the shuffled and unshuffled lists as inputs and outputs the adjacency matrix is the *core* of our model. In the experiments above, we used a graph network as the core, however other networks could be substituted. For example, transformers (Vaswani et al., 2017; Wang et al., 2018) may be used. Figure 13 shows that with two layers of transformers, the AlignNet achieves 100% accuracy.

## F    ALIGNING SEQUENCES OF SETS OF ENTITIES OVER TIME.

Figure 14 shows the accuracy with which a trained Aligner writes to memory, for different numbers of kept slots, $num\_keep\_slots$, in the input, with no further training. This is visualised in Figure 15. Each symbolic object in each slot of $U$, $S$ and $M$ is rendered as an image in a different row (or slot).

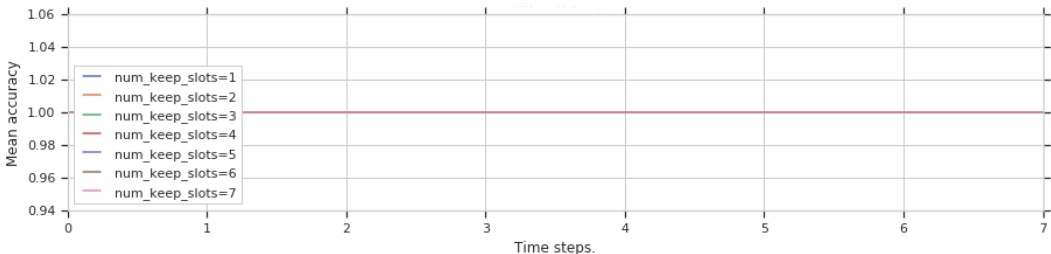

Figure 14: **Accuracy with which sequences with different numbers of kept objects are aligned.** We use the best trained Aligner from Section 4.4 and use it to align sequences – in which objects appear, disappear and reappear – with different numbers of kept objects. The model achieves 100% accuracy at each alignment step for sequences with different numbers of kept objects.

## G    ALIGNNET ARCHITECTURE AND TRAINING DETAILS.

For the results presented in this paper, the AlignNet was trained using an RMS-Prop optimiser with learning rate 2e-4, however Adam worked well too. For the graph network, we used a four layer MLP with sizes [128, 256, 512, 1024] for the edge model (Battaglia et al. (2018)) and used a three

layer MLP for the node model with sizes [32, 32, 7]. We found that the model was robust to many different MLP configurations and worked with edge models with as few as two layers and 32 nodes in each layer. However, training was faster with deeper networks and more nodes.

## H EPISODIC QUESTION ANSWERING

### H.1 EPISODIC-SORT-OF-CLEVR DATASET

The episodic-sort-of-clevr question answering dataset contains episodic sequences of objects arranged together in a *scene*, along with a question about a scene and a corresponding answer. The object shapes can be triangles or squares and each object has a unique colour. There are several types of question, all asking about the relationship between objects in the scene. For example: *'What is the colour of the farthest object from the blue object?'* and *'How many objects have the same shape as the red object?'* Questions are encoded as a binary string. Specifically, a one-hot representation of an object colour is provided, along with a binary coding of the question type ('farthest distance', 'nearest distance' or 'object count') and a binary coding of an object shape ('triangle' or 'square').

The scene is presented episodically as a sequence of observations. Each object can appear, disappear and reappear again, randomly, across the sequence of observations, just as objects in the natural world can appear, disappear and reappear as an observer changes view-point, for example. To answer a question successfully, a model must keep track of all the objects across a sequence of observations, and then answer a question about the entire scene. For example, suppose a red square appears in the first observation of a scene, a blue square appears in the second observation and in the third observation, a green triangle appears along with a repeated appearance of the red square. In order to answer a question about which object is nearest to (or furthest from) the green triangle, a model must retain a representation of the full scene in its memory, across all observations, before it can answer the question.

In the episodic-sort-of-clevr dataset, there can also be multiple scenes within an episode, each containing a different set of objects. Each question is directed towards one scene only using a unique scene identification label. Each scene can consist of a sequence of observations. These observations, each from a different scene can be interleaved amongst each-other. Therefore, in order to successfully answer a question, a model must learn to represent each scene from multiple observations, without confusing objects from across scenes before it can then answer a question about the relationship between the objects within a particular scene. As the number of scenes and observations increase, it will become increasingly difficult for many standard models to scale sufficiently to support relational question answering.

The episodic-sort-of-clevr dataset is an episodic version of the Sort-of-Clevr dataset (Santoro et al., 2017), in which all objects are presented simultaneously within a single scene, which in turn, was inspired by the Clevr question answering dataset.

### H.2 ALGINNET WITH RELATION NETWORK MODEL

We train an AlignNet together with a relation network, RN (Santoro et al., 2017), which we refer to as the AlignNet-RN, to solve the episodic-sort-of-clevr tasks. We first train the AlignNet to align objects within scenes and across observations. After training, the AlignNet may be used to write entities, in a given episode, to a memory, $M = [m_0, m_1, ...., m_N]$. At the end of an episode, when all entities have been written to memory, the memory can be treated as a set of objects, since the order is no longer important.

Next, we provide the AlignNet memory for a given episode as input to a RN. The AlignNet memory can be naturally combined with the RN, because the RN treats its inputs as a set of objects, and because the RN has in-built relational structure that is matched to the task of reasoning about relationships.

The RN receives the question $q$ and the memory, $M$, as input and is trained to produce an answer $a = f_\phi \left( \sum_{i,j} g_\psi(m_i, m_j, q) \right)$ as output. For $g_\psi$, we use a four-layer MLP with 256 units per layer and ReLu non-linearities. For $f_\phi$ we use a two-layer MLP, again with 256 units per layer and ReLu non-linearities, followed by a linear layer. We use a learning rate of $0.0001$ and a batch size of $8$. We

train all of our models using the Adam optimiser with decay rate parameters $\beta_1 = 0.9$, $\beta_2 = 0.999$ and $\epsilon = 10^8$.

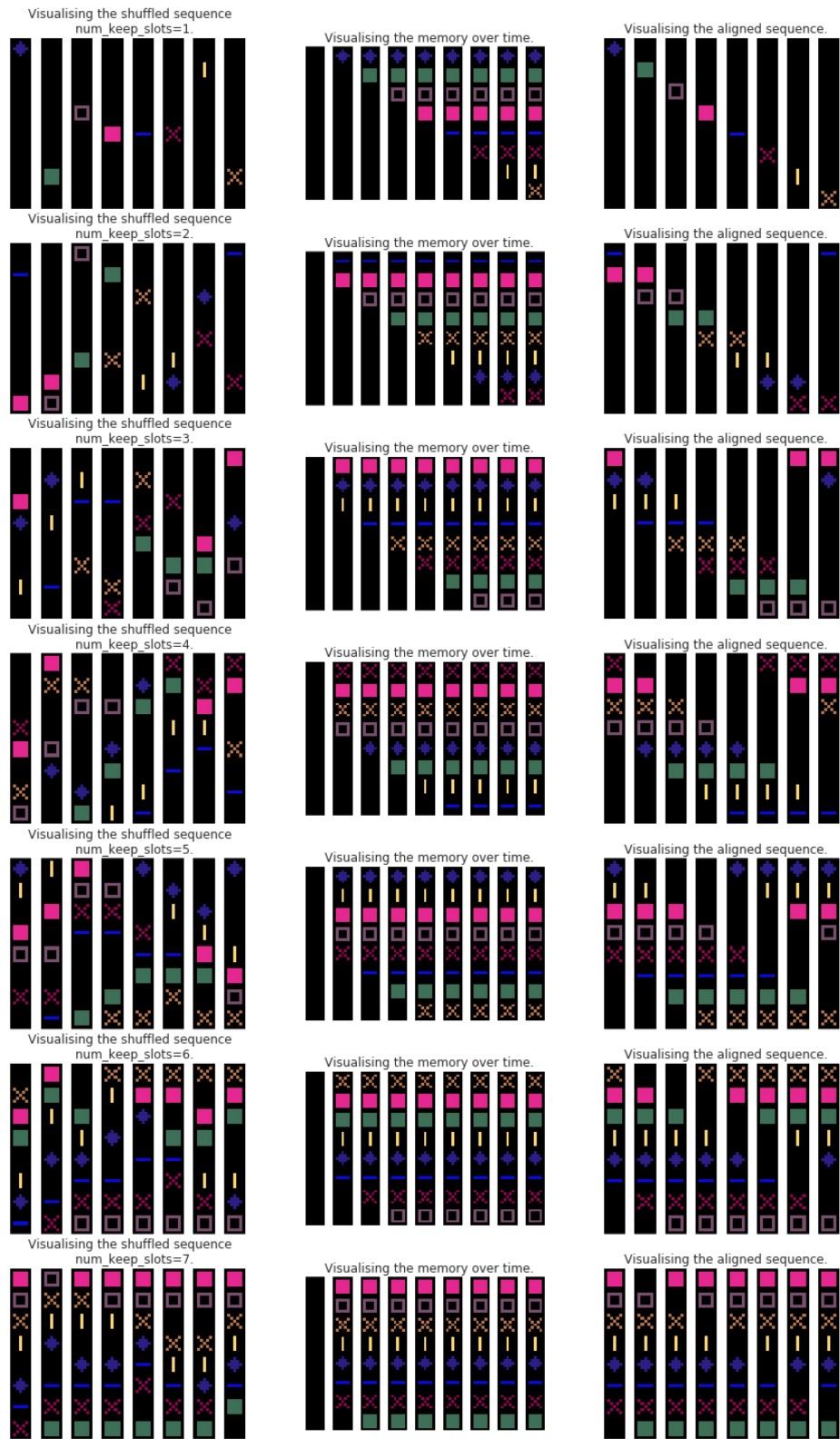

Figure 15: **Visualisations of the aligned sequences over time.** For each $num\_keep\_slots$ in the input sequence, we show how a train AlignNet, which we refer to as an Aligner, writes to memory, $M$. We also show the resulting sequence of aligned objects. We render each symbolic object representation, in each slot as an image, to show all the objects in each slot.

