# OpenReview forum: "AlignNet: Self-supervised Alignment Module"
_ICLR.cc/2020/Conference — Reject_

### Official Review · AnonReviewer2 · 2019-10-22
**Official Blind Review #2**

**Rating:** 3

**Review:**


This paper proposes AlignNet, a bipartite graph network that learns to match to sets of objects. AlignNet has a slot-wise object-based memory that associates an index with each unique object and can discover new and re-appearing objects. Experiments are conducted on a symbolic dataset.

I do not think the paper meets the acceptance threshold, and recommend for weak rejection. While the paper proposes an interesting architecture to address the alignment problem, it has noticeable flaws in its experimental designs.

First, all the experiments are conducted on toy symbolic datasets, where the alignment problem is rather easy to solve. On the other hand, real-world scenarios can be far more complicated. For example, the appearance of the same object can change due to lighting and distance, and it is unreasonable to assume that their features would remain static (apart from simple uniform noises). In addition, the paper only compares against hand-crafted similarity measures (MSE and cosine). It is unfair to compare learned methods only to hand-crafted methods. As a reasonable and fair comparison, the paper should also compare AlignNet against learned similarity measures (such as a neural network supervised with ground-truth labels for alignments).

The toy dataset and simple baselines in this paper raise doubts on whether the proposed method is applicable to more complex scenarios (such as aligning two sets of objects in natural images through their appearance features).

**Experience Assessment:**

I have read many papers in this area.

**Review Assessment: Checking Correctness Of Derivations And Theory:**

N/A

**Review Assessment: Checking Correctness Of Experiments:**

I carefully checked the experiments.

**Review Assessment: Thoroughness In Paper Reading:**

I read the paper at least twice and used my best judgement in assessing the paper.

---

> ### Author Response · Authors · 2019-11-13
> **Response to reviewer 2:**
>
> The authors are grateful for reviewer two’s interest in our work and for their thoughtful feedback.
>
> We have addressed the comments of reviewer two in detail below:
>
> R2: The alignment problem is rather easy to solve
> Response: Our paper is the first to formulate the alignment problem such that it is easy for a network to solve the problem. Before we formulated the alignment problem in this way neural networks could not be trained to align sets of objects that only partially overlapped (i.e. sets with added or dropped objects). We appreciate that this contribution may not have been clear enough in our original submission and have added the following to our list of contributions:
>
> “We are the first to formulate the alignment problem such that a neural network may learn a similarity measure for aligning entities. We propose the self-supervised AlignNet, a model capable of finding entities in one set that correspond with entities in another set, while also being able to deal with the adding and dropping of entities.”
>
> R2: It is unfair to compare learned methods only to hand-crafted methods. As a reasonable and fair comparison, the paper should also compare AlignNet against learned similarity measures.
>
> Response: We agree that it would have been nice to compare to other learned similarities, however to the best of our knowledge, we are the first to formulate the alignment problem in a way that allows neural networks to learn an implicit similarity measure and solve the alignment problem in cases where objects appear or disappear. For this reason there were no other models to quantitatively compare to. The aim of these experiments was to show that hand-crafted baseline were not sufficient and that it was necessary to train a network to solve this problem. Since we were unable to make a quantitative comparison we provided an extensive qualitative comparison to related work to demonstrate where our work sits in the field. Our work is related to both object tracking, combinatorics and memory so we qualitatively compare the AlignNet to models, from each of these fields, that most closely match our own. We also draw links to human object identification and individuation.
>
> R2: The appearance of the same object can change due to lighting and distance, and it is unreasonable to assume that their features would remain static (apart from simple uniform noises).
>
> Response: We appreciate this comment, however, we have chosen to decouple the problem of representing objects from the problem of aligning objects over time. There are already many excellent models for representing objects; for example, MONet and Iodine, but the problem of aligning objects is still an open one.

---

### Official Review · AnonReviewer3 · 2019-10-22
**Official Blind Review #3**

**Rating:** 6

**Review:**

With the assumption of object persistence and inspired by the sticky indices, this paper proposed a novel object alignment method for matching arbitrary entities in different sets. The motivation is clear and easy to follow. The simulation experiment with the symbolic dataset gives impressive results.

I like this idea and have one concern about the experiment. To be specific, all the reported results are obtained in the symbolic dataset to simulate the real-world case. Have you performed the entity alignment in the real-world data, such as the cross-lingual knowledge graphs, or cross-modal analysis? It's expected to experiment with more challenging datasets. By the way, have you released your code? This is important to an ICLR submission.


**Experience Assessment:**

I have published in this field for several years.

**Review Assessment: Checking Correctness Of Derivations And Theory:**

I assessed the sensibility of the derivations and theory.

**Review Assessment: Checking Correctness Of Experiments:**

I assessed the sensibility of the experiments.

**Review Assessment: Thoroughness In Paper Reading:**

I read the paper at least twice and used my best judgement in assessing the paper.

---

> ### Author Response · Authors · 2019-11-13
> **Response to reviewer 3:**
>
> The authors would like to thank reviewer three for their feedback and for appreciating our contributions. We agree that experiments on real-world datasets would be interesting, however the motivation for using a symbolic dataset was that we were better able to focus on the alignment problem, rather than the problem of extracting and representing entities.
>
> Your suggestion of cross-modal and cross-lingual is really interesting and definitely something we will be considering for future work, thank you.

---

### Official Review · AnonReviewer1 · 2019-10-27
**Official Blind Review #1**

**Rating:** 1

**Review:**

This paper presents a model that is able to associate objects seen in a new time step with the objects seen in previous frames and therefore consider the cases of adding new objects, dropping object no longer visible, but keeping them in memory in case they reappear in the future. The model assumes that the location of the objects is given and the only task to perform is the correct association of instances. Results shows that the proposed approach is better than hand-crafted features on different simulated tasks. Additionally the propose model is shown to help to reduce the computational cost for question answering task.

I lean to reject this paper because in my opinion the proposed method is just a set of predefined rules to train a network to be able to perform object association between a new time step and a memory. Additionally the experimental evaluation is very weak in several points (see below).

- Contribution: this approach proposes a set of rules to train a network to be able to learn the correct association  between two set of features. Additionally, the network is used together with a memory to keep track of previously seen objects. In my understanding the proposed work presents a set of hard-coded rules to train a network for tracking. However, the connection with tracking is presented only in related works (at the end of the paper) and no comparison with other tracking approach is presented. Additionally, the network model that is actually used for the experiments (graph network) is not presented in the main paper, which makes more difficult to understand how the network works for the given task.

- Experimental evaluation: Experiments are performed on a very simple, simulated environment. Objects are very simple to recognize and their location (which is often  a difficult component of the problem) is given. As the task is quite straight forward, the model obtains almost 100% on all tests. Comparisons are made only with hand-crafted features. It is quite evident that a learned similarity measure will be better than hand-crafted distances.

Authors should explain more clearly that the proposed contribution is a set of rules used to learn a distance between features in order to associate object instances.
A cleared connection with tracking should be provided from the introduction. In the evaluation there should be a comparison on multiple and more challenging tracking datasets. In this way we can compare the proposed training technique with other tracking approaches that also learn to a distance in order to associate object at different time steps.



**Experience Assessment:**

I have published one or two papers in this area.

**Review Assessment: Checking Correctness Of Derivations And Theory:**

I assessed the sensibility of the derivations and theory.

**Review Assessment: Checking Correctness Of Experiments:**

I assessed the sensibility of the experiments.

**Review Assessment: Thoroughness In Paper Reading:**

I read the paper at least twice and used my best judgement in assessing the paper.

---

> ### Author Response · Authors · 2019-11-13
> **Response to reviewer 1:**
>
> The authors would like to thank reviewer one for taking the time to read and understand our paper. However, we realise that we did not sufficiently emphasise the extent of our contribution and would like to take this opportunity to persuade the reviewer that our contribution is both original and substantial. In our paper we provide a novel framework in which a neural network is able to learn a similarity measure. The AlignNet defines a formal setting, which the reviewer refers to as “just a set of rules”, in which a network may be used to learn an implicit similarity measure. Indeed, we are the first to formulate the alignment problem in such a way that a neural network may be trained to solve the problem. Thank you for raising this concern, to make our contribution more clear we have added the following to our list of contributions in the revised paper:
>
> “We are the first to formulate the alignment problem such that a neural network may learn a similarity measure for aligning entities. We propose the self-supervised AlignNet, a model capable of finding entities in one set that correspond with entities in another set, while also being able to deal with the adding and dropping of entities.”
>
> Obtaining this formulation was non-trivial and required many iterations.
>
> We will now address the reviewers comments in more detail:
>
> R1: No comparison with other tracking approach is presented.
>
> Response: One may consider that there are two parts to multi-object tracking, one is the detection and representation of objects and the other is to align objects over time. The detection and representation of objects is a computer vision problem that has been extensively studied. In our  paper we reference several machine learning models including Tagger, MONet and Iodine which tackle the problem of object detection and representation. For this reason, we are not interested in solving the detection problem, we assume that objects have already been detected and represented. Rather, we are interested in solving the alignment problem. In particular, we consider solving the alignment problem in the very challenging, partially observable case where objects appear, disappear from view and then reappear some time later. We are the first to solve the alignment problem in this challenging, partially observable setup.
>
> Since we were unable to make a quantitative comparison we provided an extensive qualitative comparison to related work to demonstrate where our work sits in the field. Our work is related to both object tracking, combinatorics and memory so we qualitatively compare the AlignNet to models, from each of these fields, that most closely match our own. We also draw links to human object identification and individuation.
>
> R1: The network model used for experiments is not presented.
>
> Response: The AlignNet is a graph network applied to a bipartite graph. This is described in Section 3.1 as follows:
>
> “We choose to model f_{theta} as a graph network applied to a bipartite graph (Battaglia et al., 2018). Graph networks take three inputs: nodes, edges and a global variable. We represent the nodes as the concatenation of shuffled and unshuffled objects, nodes = [S, U], where every node in U is connected to every node in S by two edges, one going in each direction. There are no connections between nodes within U or S. We do not use the global variable.”
>
> We realise that we did not refer to the architecture described in appendix G from the main text and so have now added a reference to it. Thank you for bringing this to our attention.
>
>
> R1: Comparisons are made only with hand-crafted features. It is quite evident that a learned similarity measure will be better than hand-crafted distances.
>
> Response: We would like to reiterate that the AlignNet provides a novel framework in which a neural network is able to learn a similarity measure. Before this work, it was not clear how to construct a network and loss by which a similarity measure may be learned. It is for this reason that we compare to hand-crafted baselines.

---

### Decision · Program_Chairs · 2019-12-19

**Decision:**

Reject

**Comment:**

This paper proposes a network architecture which labels object with an identifier that it is trained to retain across subsequent instances of that same object.

After discussion, the reviewers agree that the approach is interesting, well-motivated and written, and novel. However, there was unanimous concern about the experimental evaluation, so the paper does not appear to be ready for publication just yet, and I am recommending rejection.